# The Effectiveness of an Annual Nationally Delivered Workplace Step Count Challenge on Changing Step Counts: Findings from Four Years of Delivery

**DOI:** 10.3390/ijerph18105140

**Published:** 2021-05-12

**Authors:** Ailsa Niven, Gemma Cathrine Ryde, Guy Wilkinson, Carl Greenwood, Trish Gorely

**Affiliations:** 1Physical Activity for Health Research Centre, Institute of Sport, PE and Health Sciences, University of Edinburgh, Holyrood Road, Edinburgh EH8 8AQ, UK; 2Division of Sport, Faculty of Health Sciences and Sport, University of Stirling, Stirling FK9 4LA, UK; gemma.ryde@stir.ac.uk (G.C.R.); guy.wilkinson@stir.ac.uk (G.W.); 3Paths for All, Kintail House, Forthside Way, Stirling FK8 1QZ, UK; carl.greenwood@pathsforall.org.uk; 4Department of Nursing and Midwifery, Centre for Health Science, University of the Highlands and Islands, Old Perth Road, Inverness IV3 5SQ, UK; trish.gorely@uhi.ac.uk

**Keywords:** physical activity, walking, workplace, occupational health

## Abstract

Paths for All’s 8-week online Workplace Step Count Challenge (SCC) is a flagship program of Scotland’s National Walking Strategy. The aim of this study was to examine changes in step counts throughout the duration of the SCC, across four years of delivery. Participants were those who registered for the 2015–2018 SCCs, and reported demographic data at registration. Participants self-reported their device-measured step count for each day of the SCC. Following data screening, mean daily steps for each week were calculated. Linear mixed models (R nlme procedure), controlling for the within subject nature of the step count measure, were used to explore changes in steps over time. Gender and age group (<45 years; ≥45 years) were entered into a subsequent model. Separate models were created for each year of the SCC and for all years combined. Participants (*n* = 10,183) were predominantly women (76.8%), aged <45 (54.6%) and ≥45 years. In general, steps increased each week compared to week 1 (*p* < 0.001), with a significant increase evident at all but seven of 28 data points. Across the four years of SCC, the increase in steps at week 8 compared to week 1 ranged from 506 to 1223 steps per day, making a substantial contribution to the recommended physical activity levels for health. There was no consistent age or gender effect. The findings provide support for the continued investment in such workplace interventions.

## 1. Introduction

The benefits of physical activity on physical and mental health outcomes are well established [1,2,3]. In order to achieve these benefits, in the UK it is recommended that throughout each week adults should at least do strength training twice a week, and 150 min of moderate-intensity aerobic activity, or 75 min of vigorous-intensity aerobic activity, or even shorter duration of very vigorous activity, or an equivalent combination of each [4]. However, large sectors of the population fail to meet these recommendations [5], and the inactivity problem has been labeled as ‘pandemic’ [6].

Walking has been identified as the simplest way many adults can achieve greater levels of moderate-intensity physical activity [7]. Evidence indicates that walking makes a key contribution to overall physical activity levels [8,9], and has health benefits [10,11], especially at a brisk pace [12]. For these reasons, and also its accessibility, walking has been identified as a ‘best buy’ for public health [13] and is an integral part of many physical activity promotion strategies (e.g., [14]), and policy [7].

The workplace is an environment in which walking can be promoted to the adult population [15,16]. Review level evidence suggests workplace physical activity interventions can be effective at increasing physical activity, although the size of the effect can vary by intervention characteristics [17,18]. There is some indication that the most promising interventions are those that target walking [18], and incorporate self-monitoring and goal setting through a wearable device [17]. A recent review by Freak–Poli et al. (2020) [19] focused exclusively on evaluating workplace pedometer interventions incorporating these promising intervention components. Findings from 14 RCTs suggested that there was evidence of increased physical activity at the end of the intervention. However, these interventions had limited effect on physical activity at least one month after the end of the program. Further, the strength of certainty in the findings of this review washindered by the low quality of the included studies.. The authors excluded other physical activity devices such as activity trackers containing accelerometers, which are now more commonly used in the general population, and only included randomized controlled trials which are less likely to be used in ‘real-world’ walking programs.

A number of both public health and private organizations deliver real world, activity tracker-based walking interventions including the Virgin Pulse (i.e., elements of the Global Corporate Challenge), Stepathlon corporate challenge, British Heart Foundation (i.e., Health at Work Pedometer Challenge), and Paths for All (i.e., Workplace Step Count Challenge). Due to their reach, if effective, such real-world interventions have the potential to have a meaningful influence on population physical activity. In relation to the Global Corporate Challenge, MacNiven and colleagues [20] reported that participants (*n* = 560) self-reported increased daily steps from 11,638 at baseline to 13,787 at the end of the 16-week intervention. However, most participants (95%) who provided pre- and post-intervention data reported that they were already meeting recommended physical activity guidelines at baseline, suggesting that the intervention was attracting physically active participants. In contrast, Freak–Poli et al. [21] reported that only 40.9% of their Global Corporate Challenge sample (*n* = 491) met the 150 min moderate-intensity physical activity recommendations at baseline. Following four months of participation, this increased by 6.5% overall, and by 27% in participants who were not reaching the recommendation at baseline. Findings from the internationally delivered annual 100-day Stepathlon [22] over three delivery years indicated that at baseline 75% of total participants (*n* = 36,562) were not achieving 10,000 steps per day, and on average improved their step count by 3519 steps/day from baseline to the end of the intervention. Although the nature of these interventions precludes the scaffolding of a robust research design to evaluate the effectiveness, collectively, these findings suggest that participation in such interventions can lead to increased physical activity over the course of the intervention for both active and inactive participants.

Paths for All’s 8-week Spring Workplace Step Count Challenge is a flagship program of Scotland’s National Walking Strategy, and has delivered an annual workplace intervention at scale (i.e., ~3000 participants per year) since 2011. Participant feedback and previous small-scale evaluation indicate this program is effective at increasing physical activity and increased step counts [23,24]; however larger scale evaluation has not yet been undertaken and such research would provide an evidence-base to support delivery. Since 2014, the Step Count Challenge has been administered through a web-based platform, on which participants log their daily steps. The use of this web-based platform offers an opportunity to capture changes in self-reported, activity tracker-measured step count across all participants who complete the Step Count Challenge, and undertake robust analysis of how step counts change across the 8-week period. Furthermore, the availability of data across consecutive years enables consideration of patterns in step count change between years, thus potentially strengthening conclusions. Finally, these data can be analyzed to consider how age and gender may influence behavior change, which will be useful to inform future intervention delivery [25]. Therefore, the aim of this research was to collate data from four years of the Step Count Challenge delivery to report who participates in the intervention, examine changes in step count during the intervention, consider step count patterns within and across years, and examine any differences in step count changes by age and gender. The findings of this study may provide an evidence base for the Step Count Challenge specifically, and contribute to the growing literature on the effectiveness of ‘real-world’ workplace walking interventions.

## 2. Materials and Methods

### 2.1. Study Design

The study was an observational design and involved secondary data analysis of four years of anonymous data from the Spring Step Count Challenge from 2015 until 2018. The Step Count Challenge takes place for 8 weeks between April and May each year. The Step Count Challenge continues to be delivered, but the data included in this study were from an inclusive four-year period. The study was approved by the Moray House School of Education and Sport Research Ethics Committee at the University of Edinburgh (Ref: 1915) and adhered to the Declaration of Helsinki.

### 2.2. Recruitment and Participants

Recruitment for the Step Count Challenge was coordinated and delivered by Paths for All. Promotional activities for the Step Count Challenge included emails to mailing lists of previous participants and general contacts; social media advertising through Facebook (including paid adverts), Twitter, Instagram, and blog posts; and taking promotional materials to events and conferences. Registration opened 2–3 months prior to the start of the challenge to allow people to promote the challenge to their workplace and for teams to sign up. Participants signed up for the challenge in teams of five employees using the online Step Count Challenge platform [26]. Some participants took part for more than one year. In participating, the participants gave informed consent to participate under Paths for All’s Step Count Challenge terms and conditions, and through this consented to the sharing of anonymous data with external partners for the purpose of evaluation.

### 2.3. Intervention

The Spring Workplace Step Count Challenge is an annual 8-week walking challenge for workplaces based in Scotland, UK, organized by a charity organization called Paths for All [26]. The Step Count Challenge is delivered through an online platform and consists of teams of five employees with a nominated team captain. The challenge costs £30 per team to enter and employees typically use their own activity tracking device or phone to record their steps. Up to 2017, pedometers could be purchased for an additional £5 per person. Each individual employee is able to enter their daily steps online, and the system includes an algorithm enabling participants to convert time spent doing popular activities, such as swimming and cycling, to steps. Using the step data entered on the first week of the Step Count Challenge, the online platform provides automatic, tailored goal setting. The goals set each week increase incrementally with the aim that by weeks seven and eight participants who have adhered will be walking an extra 5000 steps in addition to their baseline steps on five days of the week. This goal setting program is based on the incremental model of other effective walking interventions [27,28]. Additional features include a team message board, a leader board for those within the team, and also a national leader board for the collective team steps. Individuals also receive an automated congratulatory email when they achieve their best step day and week, and receive an e-mail prompt if they have not added data for more than three days. Incentives are also part of the challenge and include competitions and prizes, which have previously included photography competitions and best team name. Our own and other’s work [23,24] has shown that the Step Count Challenge incorporates a number of behavior change techniques [29] including: (i) providing information about the consequences of the behavior; (ii) goal setting; (iii) self-monitoring of behavior; (iv) feedback on performance; (v) reviewing of goal-setting; and (vi) social support; some of which are known to be effective in promoting physical activity [30] and walking specifically [31].

### 2.4. Data Analysis

Chi square analyses were undertaken to examine differences in participant characteristics by gender and age (<45; ≥45 years). Following Tudor-Locke et al. [32] the steps data were initially screened for extreme values. Any days with less than 1000 steps were deleted and any days with greater than 30,000 steps were truncated (i.e., set to 30,000 steps). For a week of data to be considered valid the participant must have recorded at least four days of steps. For each week of the challenge, the mean daily steps for that week were calculated. Linear mixed models (R nlme procedure) which control for the within subject nature of the step count measure were used to explore changes in steps over time. Models of steps by time using different repeated covariance types were conducted, and the model with the best fit was selected as the baseline model on the basis of Akaike information criterion (AIC) (model 1 in Table 2). The best fitting covariance structure was AR (1). Gender and age group (<45 years; ≥45 years) were entered into a subsequent model (model 2 in Table 3). Separate models were created for each year of the Step Count Challenge and for all years combined.

## 3. Results

Table 1 describes the participants included in the analysis for each year of the challenge. Only participants who provided both gender and age data, plus at least one week of steps data were included (86–94% of participants). The number of participants included in the analysis was lower for year 2018 due to a computer issue with recording the data. There was no difference in week one mean daily steps between those who provided sufficient data and those who did not (*p* > 0.05). Typically, around 75% of participants each year were women, and slightly more were in the <45-years-old group (approximately 55%). These differences in gender and age were significant each year (*p* < 0.05).

Changes in steps over time for each year of the Step Count Challenge and for all years combined are reported in Table 2 and displayed in Figure 1. In general, steps increased each week compared to week 1 (*p* < 0.001), with a significant increase evident at all but seven data points (i.e., week 2 in years 2015 and 2016; weeks 3, 5, 6, and 7 in 2017; week 7 in 2018). Across the four years of Step Count Challenge, the increase in steps at week 8 ranged from 506 to 1223 steps per day compared to week 1. It is worth noting that these are likely to be conservative estimates as week 1 steps were from the first week of the intervention, and therefore may not reflect a true baseline.

When age and gender were added to model (Table 3) there were no consistent 3-way or 2-way effects. There was a consistent effect for gender, with women recording fewer steps than men (*p* < 0.05, mean difference 1020.1 steps). There were inconsistent age effects. In three of the four years there was no effect for age, but in 2017 older participants (≥45 years) did more steps.

## 4. Discussion

Real-world, large-scale workplace walking interventions have the potential to facilitate meaningful population level changes in physical activity. However, limited research has examined the effectiveness of these interventions. The purpose of this study was to use data collected across four years of delivery of the Paths for All Workplace Step Count Challenge to consider who participates in the Step Count Challenge, and examine changes in step count during the intervention, consider patterns within and across years, and examine any differences in step count changes by age and gender.

Across each year of delivery, on average 75% of participants were women. In terms of age, there was a greater proportion of participants who were under 45 years of age. These findings that the Step Count Challenge is more attractive to women and those who are under 45 years are consistent with previous studies of similar programs with regards to age including the Global Corporate Challenge [20], but differs from other studies in relation to gender split [21,22]. Nevertheless, in general this finding is consistent with previous research that has reported that men are more difficult to recruit to workplace health promotion [33] and walking interventions [34]. Consequently, it may be beneficial to target men using gender-sensitized strategies [35]. However, given that women are consistently less active than men across the lifespan [36], it is encouraging that they are attracted to such interventions.

The average week 1 step count data was consistently >10,000 steps for each year, suggesting that the intervention may be attracting participants who are already active. This finding is consistent with other similar studies [20], and represents a challenge for deliverers to attract participants who are low active. However, it should be noted that the data at week one for the Step Count Challenge are not ‘true’ baseline data, as would be seen in an experimental research design. Although participants are instructed to continue their activity as normal in week 1, it is likely that merely starting the Step Count Challenge and using their activity tracker will lead to an awareness of their activity levels and therefore alter behavior and increase steps. Consideration of the pattern of change in steps at week 2 provides some support for the suggestion that steps recorded in week 1 were elevated by starting the program. Specifically, there were only relatively small increases in steps from week 1 to week 2, where the change was only significant for two out of four years, and the greatest increase was only 352 steps (2018).

Despite a potentially elevated step count at week 1 compared to a true baseline, there was strong and largely consistent evidence across the years of delivery that participants took more steps in the subsequent weeks of the challenge than in week 1. This increase from week 1 to week 8 was on average 906 steps per day by week 8, equating to 6342 steps per week. Although there is no mechanism within Step Count Challenge to collect detail on intensity of steps, based on an average cadence of 100 steps per minute for an adult walking at a moderate pace [37,38], this increase represents nine minutes per day and 63 min per week of moderate intensity activity. An increase of 63 min makes a meaningful contribution to the physical activity recommendation of 150 min of moderate intensity physical activity per week for health benefits. However, it is notable that the increase of 906 steps per day falls short of the Step Count Challenge integrated goal setting target of increasing steps by 5000 steps on five days per week from week 1 by the end of the intervention. Further, these step changes are less than the daily step count change from baseline to end of intervention reported in review level evidence (+1900) [19], by the Global Corporate Challenge (+2149) [20], Stepathlon (+3519) [22], and a previous small-scale (*n* = 20) Step Count Challenge evaluation (+1487) [23]. These differences in the other studies may at least partially be explained by the inclusion of a true baseline measurement point in these other studies, which is not elevated by the commencement of the intervention. Future Step Count Challenge research should attempt to capture pre-intervention baseline data. In terms of delivery, it should be noted that the current iteration of the Step Count Challenge has been revised to stipulate that participants aim for 6000 steps in week 1 to reduce the likelihood of an elevated step count, a ceiling effect, and potentially unachievable progression.

A more nuanced consideration of the pattern of change across the eight-week Step Count Challenge is helpful to inform any targeted intervention; however, this consideration does not highlight any consistent pattern other than that there was a consistent increase on step count from week 1 across the eight weeks. Out of 28 data points (four years x seven weeks of data), there were only seven points where there was not a significant increase in steps compared with week 1. As noted above, there was no significant increase from week 1 to week 2 for two of the four years, possibly due to elevated week 1 steps. There were also three data points for 2017 (weeks 3, 5, and 7) where there was not a significant difference, and one data point in week 6 where there was a significant reduction from week 1. It is likely that these findings are due to the high week 1 data in 2017, which was the highest out of the four years. For both 2017 and 2018, there was no difference between week 1 and week 7; although it is premature to draw firm conclusions from these findings, they could suggest that engagement may dip at this point and a targeted ‘push’ may be beneficial to re-engage participants.

Consideration of the effects of gender and age on an intervention is helpful in order to inform future intervention development to target or sensitize the intervention to specific groups. Women consistently recorded fewer steps than men, consistent with population level data that highlights women as less active [36]. Nevertheless, there were no gender differences in the response to the intervention as both men and women exhibited the same upward increase from baseline across each week. This lack of gender differences is consistent with previous studies [20,22], and is encouraging, suggesting that the intervention can be effective for both men and women who sign up. There was no indication that age, nor age X gender had an impact on the effectiveness of the intervention, suggesting the Step Count Challenge can be beneficial for all men and women participants over and under 45 years of age.

A strength of this study is that we were able to use data from multiple iterations of the intervention, enabling a pooling of a dataset from a large number of participants, and consideration of patterns across years to reinforce findings. Limitations include the use of a range of self-reported activity tracker data and lack of consideration of cadence of steps, both of which may challenge the validity of the data. Further, the lack of a true baseline and no control group limits the potential to draw definitive conclusions, and future research should aim to address these, although there are challenges to implementing such a study design within ‘real-world’ interventions. Future research could consider how best to attract more men and low active participants to the intervention. Previous research suggests that longer term behavior change is minimal with workplace pedometer interventions [19], so further Step Count Challenge research that collects data beyond the eight weeks of the challenge would be useful to understand the potential longer term behavior change benefits of participation.

## 5. Conclusions

The Workplace Step Count Challenge consistently attracts a large number of participants who report an average increase in steps from week 1 by week 8 equivalent to 63 min per week. The intervention attracted a greater proportion of women and <45-year-olds, but these characteristics did not influence the effectiveness of the intervention. The findings of this study add further to the evidence base on the effectiveness of such workplace interventions to promote physical activity, and specifically provide support for the continued investment in the Step Count Challenge.

## Figures and Tables

**Figure 1 ijerph-18-05140-f001:**
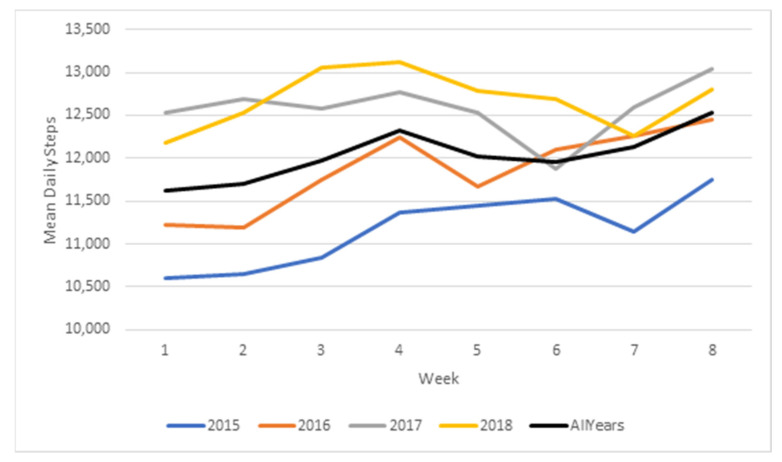
Change in step count over 8 weeks of Step Count Challenge by year and all years.

**Table 1 ijerph-18-05140-t001:** Participant breakdown by Step Count Challenge year, gender, and age.

SCC Year		Gender (*n* and % of Total)		Age Group (*n* and % of Total)	
	*n*	Men	Women	<45 Years	≥45 Years
2015	2153	476 (22.1%)	1677 ^a^ (77.9%)	1206 ^b^ (56.0%)	947 (44.0%)
2016	3666	882 (24.1%)	2784 ^a^ (75.9%)	1976 ^b^ (53.9%)	1690 (46.4%)
2017	3375	746 (22.1%)	2629 ^a^ (77.9%)	1842 ^b^ (54.6%)	1533 (45.4%)
2018	989	254 (25.7%)	735 ^a^ (74.3%)	534 ^b^ (54.0%)	455 (46.0%)
Overall	10,183	2358 (23.2%)	7825 ^a^ (76.8%)	5558 ^b^ (54.6)	4625 (45.4%)

^a^ Significantly more than men (*p* < 0.05); ^b^ significantly more than ≥45 (*p* < 0.05); SCC = Step Count Challenge.

**Table 2 ijerph-18-05140-t002:** Model 1: Change in mean (standard error) daily step count over time relative to week 1.

Week	Step Count Challenge Year				
	2015 (*n* = 2153)	2016 (*n* = 3666)	2017 (*n* = 3375)	2018 (*n* = 989) ^a^	All Years Combined (*n* = 10,183)
(Intercept)	10,604.0 (92.0) *	11,219.9 (73.1) *	12,533.8 (79.8) *	12,179.7 (142.4) *	11,620.0 (44.8) *
Week 2	53.4 (57.8)	−23.2 (44.7)	150.5 (45.7) *	352.4 (87.4) *	88.0 (26.8) *
Week 3	242.5 (67.5) *	530.9 (51.5) *	51.9 (53.1)	880.8 (99.7) *	345.2 (31.1) *
Week 4	763.5 (70.8) *	1028.2 (53.8) *	230.3 (55.7) *	938.5 (103.5) *	697.5 (32.6) *
Week 5	849.4 (72.4) *	450.1 (55.1) *	1.8 (57.0)	605.6 (105.1) *	400.1 (33.3) *
Week 6	926.7 (73.6) *	875.2 (55.8) *	−651.2 (57.8) *	505.7 (106.0) *	339.5 (33.7) *
Week 7	544.4 (74.6) *	1041.0 (56.5) *	59.6 (58.6)	75.6 (107.4)	513.0 (34.2) *
Week 8	1140.2 (76.3) *	1223.4 (57.7) *	506.2 (59.9) *	621.1 (109.7) *	906.1 (34.9) *
AIC	284,272.1	485,210.7	455,210.4	135,504.7	1,361,550

* *p* < 0.001; ^a^ smaller sample size was due to computer download issue.

**Table 3 ijerph-18-05140-t003:** Model 2: Gender and age effects on change in step count (mean and standard error) over time relative to week.

AR(1)	SCC2015	SCC2016	SCC2017	SCC2018	All Years Combined
(Intercept)	11,564.3 (272.6) ***	11,843.5 (196.1) ***	12,929.6 (219.4) ***	12,609.7 (373.4) ***	12,235.1 (123.2) ***
Week 2	22.3 (173.7)	−126.4 (120.8)	75.7 (126.4)	765.4 (233.4) ***	65.5 (74.4)
Week 3	−44.5 (204.9)	356.0 (139.1) **	109.0 (146.8)	1419.5 (264.6) ***	319.0 (86.3) ***
Week 4	568.2 (215.9) ***	1029.6 (145.8) ***	189.1 (153.8)	1145.0 (276.1) ***	677.0 (90.6) ***
Week 5	727.1 (219.9) ***	367.3 (149.3) **	−14.3 (157.9)	613.7 (278.8) **	330.8 (92.7) ***
Week 6	946.2 (223.0) ***	826.6 (151.9) ***	−863.8 (160.5) ***	959.1 (280.3) ***	293.0 (94.1) ***
Week 7	453.9 (226.5) **	898.9 (154.6) ***	65.7 (163.1)	137.3 (284.3)	451.2 (95.7) ***
Week 8	1412.1 (232.2) ***	787.1 (157.2) ***	307.8 (167.5) *	663.6 (288.0) **	719.4 (97.8) ***
Gender	−1357.4 (304.9) ***	−749.9 (227.3) ***	−866.6 (251.7) ***	−1106.4 (435.7) **	−983.4 (141.3) ***
Age Group	−109.1 (388.8)	277.8 (300.4)	826.4 (344.5) **	863.6 (562.5)	367.3 (186.9) **
Week 2 × Gender	3.2 (194.1)	102.2 (139.9)	144.5 (145.0)	−355.3 (271.6)	44.2 (85.3)
Week 3 × Gender	283.3 (228.5)	114.0 (161.1)	10.4 (168.5)	−466.4 (308.7)	27.8 (99.0)
Week 4 × Gender	130.1 (240.6)	−7.9 (168.8)	40.6 (176.6)	−21.4 (321.8)	17.2 (103.9)
Week 5 × Gender	87.1 (245.3)	41.1 (172.9)	−40.0 (181.3)	108.4 (325.6)	44.4 (106.3)
Week 6 × Gender	−146.1 (249.0)	−26.6 (175.8)	258.8 (184.1)	−329.0 (327.7)	20.3 (107.9)
Week 7 × Gender	17.18 (253.0)	52.3 (178.9)	−61.6 (187.2)	−2.8 (332.3)	−2.8 (109.7)
Week 8 × Gender	−472.8 (259.5) *	530.1 (182.2) ***	205.2 (1912.0)	195.4 (337.7)	200.5 (112.1) *
Week 2 × Age Group	133.5 (246.2)	−32.4 (184.8)	205.9 (197.8)	−424.6 (348.1)	38.9 (112.5)
Week 3 × Age Group	593.4 (287.9) **	432.3 (212.2) **	−194.8 (229.9)	−732.0 (396.0) *	144.4 (130.2)
Week 4 × Age Group	147.9 (302.7)	10.8 (222.2)	134.8 (240.3)	−320.3 (411.3)	53.6 (136.4)
Week 5 × Age Group	116.1 (308.0)	−129.7 (226.9)	67.7 (246.3)	−202.7 (417.0)	12.5 (139.3)
Week 6 × Age Group	211.2 (311.9)	41.3 (230.1)	93.0 (249.8)	−785.0 (421.0) *	82.1 (141.2)
Week 7 × Age Group	260.0 (316.8)	74.2 (233.1)	−86.6 (252.8)	−317.8 (425.0)	41.4 (143.1)
Week 8 × Age Group	59.2 (324.3)	230.8 (236.6)	142.6 (258.5)	−543.9 (433.3)	135.7 (145.9)
Gender × Age Group	438.3 (441.5)	−508.6 (343.8)	−263.9 (388.8)	−13.5 (651.5)	−73.8 (212.9)
Week 2 × Gender × Age Group	−91.1 (279.5)	112.7 (211.3)	−361.2 (223.3)	139.3 (402.8)	−82.4 (128.1)
Week 3 × Gender × Age Group	−593.5 (326.7) *	−311.5 (242.9)	63.5 (259.6)	421.2 (458.7)	−173.0 (148.4)
Week 4 × Gender × Age Group	81.8 (343.3)	0.5 (254.3)	−144.8 (271.6)	−119.4 (476.3)	−49.5 (155.4)
Week 5 × Gender × Age Group	6.8 (349.7)	310.6 (259.7)	41.0 (278.2)	13.6 (483.2)	80.7 (158.8)
Week 6 × Gender × Age Group	−7.5 (354.2)	138.5 (263.3)	−88.6 (282.2)	443.1 (487.6)	−21.1 (160.9)
Week 7 × Gender × Age Group	−120.2 (359.7)	185.9 (266.7)	216.5 (285.6)	248.8 (492.9)	121.4 (163.0)
Week 8 × Gender × Age Group	197.4 (368.1)	−200.2 (271.2)	−79.1 (291.9)	191.3 (502.9)	−85.4 (166.3)
AIC	283,962.4	484,908.4	454,911.4	135,202.6	1,361,200

* *p* < 0.05; ** *p* < 0.01; *** *p* < 0.001; SCC = Step Count Challenge.

## Data Availability

The data employed in this study were used under license from Paths for All. Participants in the Step Count Challenge agreed to Paths for All’s Step Count Challenge terms and conditions, which included agreement that anonymous data could be shared with external partners for the purpose of evaluation and reporting. Given this level of consent, the data cannot be shared publicly. However, data may be available upon reasonable request from Paths for All.

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
