# Peer review of "The Effectiveness of an Annual Nationally Delivered Workplace Step Count Challenge on Changing Step Counts: Findings from Four Years of Delivery"

_ijerph, 2021, doi:10.3390/ijerph18105140_

Round 1
Reviewer 1 Report
I find it very interesting that the introduction talks about other studies with the same purpose as this one, because it shows that this is something of interest to society and something that should be further researched.
Even so, I miss that the authors develop more on the subject of sedentary lifestyles and the importance of PA in the daily life of the population. They only talk about this specifically in the first paragraph of the article.
The data they analyse was collected between 2015 and 2018. Did the study fully end in 2018, and have participants been followed up to see if they have maintained their habits over time?
The article talks about this study as an intervention. I believe that the main thing that this study brings to the population is motivation to do PA, in a simple and affordable way for everyone. This motivation is achieved through prizes, competitions, rankings... Do the authors think that without these incentives so many people would have signed up, or that this is a secondary issue?
Are participants adhering to the practice of PA, or are they going to stop doing PA once the follow-up is over?
I think the conclusions are too brief
The article in general is very interesting, because it provides data that justifies the need to create this type of programme to motivate the population to do PA. I think the authors should focus more on whether true adherence to PA is achieved, which should be the main aim of all these studies.
Reviewer 2 Report
Dear author,
This article entitled “The effectiveness of an annual nationally delivered Workplace Step Count Challenge on changing step counts across four years of delivery” presents an 8-week workplace step count as Scotland's National Walking Strategy. It aims to examine changes in step count in 10,183 participants, 76% of whom were women.
It is original and offers extensive knowledge and contribution to health by counting steps in health and physical activity. However, there is much room for improvement in the theoretical conceptualization to better understand this knowledge. Being necessary to contribute studies and concepts that better integrate its publication. It is necessary to improve the contribution of the literature, where it presents publications and conceptualizations. The contribution is significant, with a clear objective.
However I present here my suggestions for improvement:-In the abstract as in the introduction, conceptualize from what perspective you speak, of older adults who work with an age of ???
-In the introduction = I see little points. That is, he comments on some publications that are interventions performed with pedometers. It also indicates that the web-based platform can quote steps for 8 weeks, and that it will difference over 4 years of delivering the step count and see changes by age and gender. The introduction is weak, you conceptualize why you want to do this and in what type of sample, put studies with the same type of sample. He tells me about physical activity and a platform, but I need him to merge it with studies similar to his sample and justify why he is doing this study. What is the importance, what is the problem to be solved, what is happening.
-In recruitment = indicate what was it, an online survey conducted with what platform? Here describe your sample, how many participants, age range, what is the minimum age, and the maximum average, what characteristics do these sociodemographic participants have, talk more about your sample.
-In Intervention = if you use your mobile device to record the steps, how do you guarantee anonymity? And further describe your evaluation instrument, which evaluates how long it takes to answer it, what dimensions / scales it has, reliability index, validity, Cronbach's alpha, describe your instrument with all the characteristics.
-In the last paragraph of the intervention you indicate that your work incorporates a series of behavior change techniques, can you specify evidence of this or how you do it? It also indicates that it gives social support, can you specify how?
-What happens at work, why do you recruit people at work, what is your interest, and why do you answer more women than men in your recruitment?
- You contribute what practical implications to society this work has, what function this work has, what it contributes.
- -Please contribute what scientific implications this work has, what it means for the scientific world, how it can help or advance with this study and why.
- Very important = in the introduction better conceptualize your research, introduce studying the workplace and physical activity and justify your study.
- -Conclusion = improve the contribution by indicating what your work contributes, what implications are derived from it to society and science
-
-It is necessary to improve the references in an appropriate format.
Best regards,
Round 2
Reviewer 2 Report
Estimated author,
The changes incorporated have substantially improved the publication that presents.
Best regards,